# Advances in the Treatment of Enterovirus-D68 and Rhinovirus Respiratory Infections

**DOI:** 10.3390/idr17030061

**Published:** 2025-06-01

**Authors:** Vonintsoa L. Rahajamanana, Mathieu Thériault, Henintsoa Rabezanahary, Yesmine G. Sahnoun, Maria Christina Mallet, Sandra Isabel, Sylvie Trottier, Mariana Baz

**Affiliations:** 1Axe Maladies Infectieuses et Immunitaires, Centre de Recherche du CHU de Québec-Université Laval, Quebec City, QC G1V 0E8, Canada; vonintsoa-lalaina.rahajamanana@crchudequebec.ulaval.ca (V.L.R.); mathieu.theriault@crchudequebec.ulaval.ca (M.T.); henintsoa.rabezanaha@crchudequebec.ulaval.ca (H.R.); yesmine.sahnoun@crchudequebec.ulaval.ca (Y.G.S.); maria-christina.mallet@crchudequebec.ulaval.ca (M.C.M.); sandra.isabel@crchudequebec.ulaval.ca (S.I.); sylvie.trottier@crchudequebec.ulaval.ca (S.T.); 2Centre de Recherche en Infectiologie, Université Laval, Quebec City, QC G1V 0E8, Canada; 3Département de Microbiologie-Infectiologie et d’Immunologie, Faculté de Médecine, Université Laval, Quebec City, QC G1V 0A6, Canada; 4Département de Pédiatrie, Faculté de Médecine, Université Laval, Québec City, QC G1V 0A6, Canada

**Keywords:** rhinovirus, enterovirus, EV-D68, respiratory infection, antivirals

## Abstract

Background/Objectives: Enterovirus-D68 (EV-D68) and rhinoviruses are major contributors to respiratory illnesses in children, presenting a spectrum of clinical manifestations ranging from asymptomatic cases to severe lower respiratory tract infections. No specific antiviral treatments are currently approved for these viruses. Method: We conducted a comprehensive literature review of antiviral agents investigated for EV-D68 and rhinovirus infections. Results: Several antiviral candidates are under investigation, each targeting distinct stages of the viral replicative cycle. Capsid-binding agents and monoclonal antibodies prevent viral attachment by blocking receptor-virus interactions. Inhibitors of viral replication proteins disrupt polyprotein processing and replication organelle biogenesis by targeting non-structural viral proteins. Host factor inhibitors impair viral attachment, replication organelle formation, or RNA replication by interfering with critical host pathways. Conclusions: While no specific antivirals are yet approved for EV-D68 and rhinovirus infections, emerging therapeutic candidates offer potential avenues for treatment. Continued preclinical and clinical investigation will be essential to validate these approaches and expand the available options for affected patients.

## 1. Background

Enterovirus (EV) and rhinovirus (RV) species commonly infect humans. They cause a wide range of clinical manifestations across all age groups: asymptomatic, upper and lower respiratory infections, aseptic meningitis, viral encephalitis, myopericarditis, acute flaccid myelitis (AFM), hand, foot and mouth disease, herpangina, viral conjunctivitis, to name a few [1,2]. They impose an important socioeconomic burden, resulting in significant healthcare expenses and productivity losses [3].

The name enterovirus comes from the Greek word “*enteron*”, meaning intestine, and the latin word “*virus*” meaning poison [4,5]. While traditionally linked to enteric transmission, certain EVs, such as EV-D68, primarily spread through respiratory routes [6]. In 2014, EV-D68 triggered an outbreak of severe lower respiratory tract infections across North America, predominantly affecting children. This outbreak was also associated with AFM, a serious neurological complication [7,8]. The increasing number of outbreaks since 2014 suggests a high potential for this virus to emerge as a significant human pathogen [7,9].

Rhinovirus, whose prefix comes from the Greek word “*rhis*”, meaning nose, is a group of viruses linked to respiratory illnesses [10]. While widely recognized as the primary cause of the common cold, their role in severe illnesses has often been overlooked [11]. In adults, growing evidence over the past few decades has linked RVs to asthma and chronic obstructive pulmonary disease exacerbations [12,13,14,15,16,17,18]. In children with underlying conditions such as prematurity, heart disease, or metabolic disorders, RV infections may result in severe outcomes leading to higher hospitalization rates and requiring intensive care [19,20].

A deeper understanding of these viruses is essential for developing effective preventive and therapeutic strategies, particularly for vulnerable populations, and to improve outbreak preparedness. This review provides a comprehensive overview of developing antiviral agents for EV-D68 and RVs, examining both historically studied antiviral approaches and those currently under investigation.

## 2. Enterovirus Classification

EVs and RVs are classified within the singular genus *Enterovirus*, under the *Picornaviridae* family, as confirmed by genomic and antigenic analyses [21]. Historically, EVs were classified into polioviruses, coxsackieviruses (A/B), and echoviruses serotypes defined by antigenic properties observed through antibody neutralization tests. However, overlapping biological properties led to a revised system using consecutive numbering supported by genotyping (e.g., EV-A71, EV-D68) [22]. Genotype classification relies on the genetic sequence analysis of the protein VP1, a key component of the viral capsid region [23]. Classification now prioritizes genetic lineage over serogroups and genotyping of VP1 has become the gold standard for EV classification, providing a more robust framework for understanding viral diversity and evolution.

At present, the *Enterovirus* genus comprises 15 species: 12 EV species (A–L) and 3 RV species (A–C). Of these, 7 species are human pathogens, including EV A–D and RV A–C (Figure 1) [4]. EV-D68 is now part of the EV-D species, which includes five genotypes (EV-D68, EV-D70, EV-D94, EV-D111, and EV-D120), each linked to distinct symptoms [24].

## 3. Virological Characteristics

EVs are positive-sense single-stranded RNA viruses with a genome of approximately 7500 nucleotides [25]. The genome is packaged in small icosahedral capsids measuring approximately 30 nanometers in diameter, composed of 60 copies of four proteins: VP1, VP2, VP3, and VP4 [26]. These proteins are organized into subunits, with VP1-VP3 positioned on the outside and VP4 on the inside, joining at the apexes to create a pattern of alternating trimer in a pentamer symmetry (Figure 2).

Next to the pentavalent apex, a canyon, a circular depression encircling the fivefold axis of symmetry on the virion surface, is thought to contribute significantly to receptor binding [27]. At the base of the canyon, the hydrophobic pocket of each VP1 subunit houses a host-derived lipid “pocket factor”, target of some antiviral molecules [26].

The viral genome consists of three segments (Figure 3): a 700–825 nucleotide long 5′ untranslated region (UTR), a single open reading frame (ORF) that encodes a viral polyprotein, and a 75–100 nucleotide long 3′ UTR. This polyprotein is cleaved by autocatalysis to produce four structural proteins (VP1–VP4) and seven non-structural proteins (2A, 2B, 2C, 3A, 3B, 3C, and 3D) [28]. The 5′ UTR contains the internal ribosome entry site (IRES), which is essential to recruit ribosomes for the initiation of translation [29]. EVs and RVs exhibit high mutation rates and frequent recombination, particularly in the 5′UTR-capsid junction and at the beginning of the P2 region, contributing to their genetic diversity and evolution [30].

## 4. Replicative Cycle

The replication of the *Enterovirus* genus occurs in the cytoplasm and involves several key steps (Figure 4). The cycle begins with viral entry, during which the virus binds to specific cellular receptors and is internalized [31]. Viral uncoating is triggered by the acidic environment of the endosome, where a lower pH causes the release of the viral genome into the cytoplasm [32]. The viral RNA genome is covalently linked to VPg (3B), a viral protein that acts as both a primer for replication and a facilitator of translation. The viral RNA is subsequently translated into a single polyprotein [28,33]. This process occurs as the virus hijacks the host cell’s ribosomes, which bind to the viral IRES in the 5′-UTR to initiate protein translation. The polyprotein is cleaved by the viral proteases 2A, 3C, and its precursor form 3CD into four structural proteins (VP1–VP4) and seven non-structural replication proteins (2A, 2B, 2C, 3A, 3B, 3C, and 3D) [21,34,35].

A crucial aspect of *Enterovirus* replication is the formation of specialized membrane structures called replication organelles (ROs), which are derived from the endoplasmic reticulum and Golgi apparatus, with viral proteins 2B, 2C, and 3A playing key roles in their biogenesis [33,36]. Replication of viral RNA occurs within these ROs, where a favorable lipid environment is created by the viral proteins 2B, 2C, and 3A, assisted by host proteins such as acyl-CoA-binding domain-containing protein (ACBD3), phosphatidylinositol 4-kinase-IIIβ (PI4KIIIβ), oxysterol-binding protein (OSBP), and OSBP-related protein 4 (ORP4) [37]. Replication begins with the transcription of the genomic (+) RNA into (−) RNA by the RNA-dependent RNA polymerase (3Dpol), and the (−) RNA then serves as a template for the synthesis of new (+) RNA [21]. Throughout this process, the virus exploits various host cell pathways and factors. The secretory pathway and autophagy are hijacked to support RO formation [37]. Lipid metabolism is altered, with lipid droplets potentially serving as lipid sources for RO proliferation [33,37]. Host factors such as heat shock proteins (e.g., HSPA9) are recruited to assist in viral protein folding and stabilization [28].

The replication cycle concludes with the assembly of newly synthesized viral RNA into capsids formed by structural proteins (VP1, VP2, VP3, and VP4). This process involves the encapsidation of RNA into capsid proteins, which self-organize into protomers and pentamers, followed by maturation in cellular structures called autophagosomes [38]. Finally, mature virions are released from the host cell, either through exocytosis, budding, or cell lysis [39].

The virus replication cycle generates virions within a few hours of the initial infection, with viral RNA replication beginning 2–3 h post-infection and translation starting shortly after at 3–4 h [28,40]. Most viral proteins reach high levels between 4 and 7 h post-infection, and new virions are typically released within 8–10 h, completing the efficient replication cycle of enteroviruses [41,42]. This efficient process, coupled with the virus’s ability to manipulate host cell machinery, contributes to the pathogenicity and rapid dissemination of *Enterovirus* infections.

## 5. Current Strategies in *Enterovirus* Antiviral Research

The biodiverse *Enterovirus* species pose unique challenges due to their ability to mutate rapidly, evade immune responses, and cause widespread outbreaks. EV-D68 exemplifies these concerns, having caused a large epidemic in North America in 2014, with concurrent outbreaks and sporadic cases also reported in Europe, Asia, Australia, and other regions, highlighting its global public health relevance [43,44,45,46,47,48,49]. Epidemiological surveillance reveals a pronounced biennial cycle of incidence peaks during summer and fall, leading to waves of medical consultations [50,51,52].

Currently, no approved antiviral treatment or vaccine exists for EV or RV infections, leaving supportive care as the primary management strategy [9,44,50,53,54,55]. Developing effective antiviral therapies requires a deep understanding of viral biology, host interactions, and infection mechanisms. This process depends on creating accurate experimental models and designing treatments that are both broad-spectrum and adaptable to viral evolution.

### 5.1. Experimental Models

Accurate experimental models are essential for studying *Enterovirus* because they provide insights into viral pathogenesis, host immune responses, and therapeutic efficacy that cannot be fully replicated otherwise. Understanding how these viruses cause disease requires models that mimic human infection. They provide the means to identify viral tropism, replication mechanisms, and host factors contributing to disease severity. Additionally, animal models are critical for testing antiviral drugs and vaccines, ensuring their safety and efficacy before clinical trials.

In this regard, cell cultures are the primary in vitro models for studying viral activity, pathogenesis, and host interactions [56,57,58]. Human organoids, such as lung organoids, provide human-relevant systems for modelling infections and offering insights into viral tropism and replication in tissue-specific contexts [59,60]. Animal models are then used for studying the infection within a complex organism. Mouse models for rhinovirus (RV) infection have historically been challenging due to species-specific receptor differences [61]. However, recent advances have enabled the development of more useful models. Transgenic mice (*Mus musculus*) (e.g., ICAM-1) and neonatal mice are widely used to replicate human-like symptoms, including neurological and systemic manifestations, and to evaluate vaccines [62,63]. Rodent models, particularly mice, have been extensively used to study *Enterovirus* infection, viral-induced exacerbation of lung diseases, and for testing antivirals and vaccines [63,64,65,66,67]. Despite their high cost, ferrets (*Mustela putorius furo*) and non-human primates are also used due to their close genetic and physiological similarity to humans [68,69,70]. Together, these models offer comprehensive frameworks for understanding *Enterovirus* pathogenesis, testing antivirals and developing vaccines.

### 5.2. Broad-Spectrum and Adaptable Treatments

Broad-spectrum antiviral approaches aim to target a wide range of viruses by interfering with essential viral or host processes required for replication, rather than focusing on a single virus or viral family [71]. These strategies include both direct-acting antivirals and host-targeted therapies [72]. Examples of the former include nucleoside analogues like ribavirin, which inhibit viral polymerases across multiple RNA and DNA viruses, and favipiravir, which targets the RNA-dependent RNA polymerase of various RNA viruses [73,74]. Host-directed approaches include type I and type III interferons (IFNs), which stimulate the innate immune response by binding to their specific receptors and activating the JAK-STAT signaling pathway. This leads to the expression of hundreds of interferon-stimulated genes (ISGs) that induce antiviral states in infected and neighboring cells [75].

To design broad-spectrum and evolutionarily adaptable treatments for *Enterovirus* species, researchers can focus on targeting highly conserved viral proteins and host factors essential for viral replication. For example, the non-structural protein 2C of *Enterovirus* can be a potential target due to its conserved role in viral replication and assembly [76]. Lethal mutagenesis uses nucleoside analogs to elevate the viral mutation rate, exploiting viral evolution to push the virus into error catastrophe, a state of irrecoverable genetic damage that renders it nonviable [77]. Another strategy involves host-targeted therapies, such as inhibiting PI4KIIIβ, a cellular enzyme critical for forming enteroviral replication organelles. This approach reduces the risk of resistance, as it targets host machinery rather than viral proteins, which are more prone to mutation. Combination therapies that pair antivirals with innate immune stimulators have shown synergistic effects in inhibiting *Enterovirus* replication. For instance, the combination of gemcitabine with ribavirin has demonstrated synergistic efficacy in reducing replication in cellular models of EV-A71 and CVB3 (EV-B) infections [78]. These last two drugs are also an example of drug repurposing, which involves leveraging existing compounds developed for other diseases. This approach has emerged as an efficient strategy to accelerate antiviral drug development since their safety and pharmacokinetics have already been evaluated [35]. Overall, these strategies combined with evolutionary insights and host–virus interaction studies provide a robust framework for developing adaptable, broad-spectrum *Enterovirus* treatments.

## 6. Emerging Antiviral Candidates for RV-D68 and RVs

Antiviral compounds aim to disrupt virus replication and spread. They typically target either viral proteins or essential host factors exploited by the virus (Figure 5). These compounds are classified based on their target proteins, as shown in Table 1.

### 6.1. Capsid Binders

Capsid binder molecules target the viral capsid of *Enterovirus* to disrupt viral entry and uncoating. They induce rigidity in the capsid by occupying the hydrophobic pocket and altering its pocket factor, enhancing its stability. This prevents conformational changes in the capsid protein, inhibits receptor binding, and blocks viral protein interactions with the host, ultimately preventing the release of the viral genome into the host cell.

#### 6.1.1. Pleconaril

Pleconaril targets viral VP1 protein and prevents virus entry into host cells. It was originally developed for the treatment of EV and RV respiratory infections and has broad activity against these viruses, including EV-D68 in vitro [34,88,123,124]. In clinical trials for the treatment of upper respiratory tract infection caused by RV, patients who received pleconaril experienced an early and sustained reduction in symptom severity, as well as a one-day decrease in disease duration [80]. Despite these positive antiviral results, pleconaril has not been approved by the Food and Drug Administrations (FDA) due to safety profile concerns [81,125]. Pleconaril induces cytochrome P-450 3A enzymes, which can interact with other medications. Notably, the antiviral reduces the effectiveness of oral contraceptives by decreasing plasma levels of ethinyl estradiol, which the P-450 3A enzymes metabolize [80]. The combination of pleconaril and oral contraceptives is also associated with menstrual irregularities, including breakthrough bleeding and spotting.

#### 6.1.2. Pirodavir and Vapendavir

Pirodavir is a capsid-binding substituted phenoxy-pyridazinamine with potent in vitro activity against most RV serotypes. This compound has demonstrated significant antiviral effects in controlled trials assessing the efficacy of intranasal pirodavir in experimentally induced RV infection of susceptible volunteers. However, it was associated with higher rates of nasal dryness, blood in mucus, or unpleasant taste on several study days [83]. Vapendavir, a pirodavir derivative, previously referred to as BTA-798, significantly reduces asthma exacerbations caused by RVs [89]. However, pirodavir and vapendavir did not show antiviral effectiveness against EV-D68 in vitro and in vivo in both respiratory and neurological infection models using AG129 mice [34,100,109]. Although both drugs remain experimental, results from a phase II clinical trial of vapendavir are expected in summer 2025 [83,89,90].

#### 6.1.3. Pocapavir

Pocapavir (V-073), an investigational capsid-binding antiviral showed significant activity against poliovirus in preclinical studies. However, in vitro testing against EV-D68 revealed a lack of effectiveness [100]. Although one study suggested potential anti-RV activity, supporting data remain limited [86]. Some cases do report clinical improvement when used for treatment of severe neonatal enteroviral sepsis and CVB3-induced myocarditis [85,126,127]. It is also available for emergency treatment of severe EV-B infections [84]. More research is needed to assess its efficacy against RV and EV-D68.

#### 6.1.4. R856932

In 2019, the tetrazole-based compound R856932 was identified as a potent antiviral agent against EV-D68 in vitro. It demonstrated high efficacy against various contemporary EV-D68 strains in the rhabdomyosarcoma (RD) cells model [88].

#### 6.1.5. Quinoline Derivative Compound 19

Compound 19, a quinoline derivative still in the preclinical stage, has shown significant EV-D68 antiviral activity. This compound exhibited broad-spectrum antiviral effects in vitro, with EC50 values ranging from 0.05 to 0.10 μM against various clinical EV-D68 strains as well as the prototype strain in RD cell lines. Its antiviral activity is believed to result from the 1,2,4-oxadiazole modification, which enhances binding to the hydrophobic pocket of the VP1 protein. Additionally, it showed good pharmacokinetic properties, including acceptable bioavailability in rat models [87].

### 6.2. Viral Replication Protein Inhibitors

Viral replication protein inhibitors disrupt key processes such as viral RNA synthesis, protein cleavage, and genome assembly, thereby preventing the production of infectious viral particles [128].

#### 6.2.1. Telaprevir—2A Protease Inhibitor

Telaprevir is an antiviral originally developed to treat Hepatitis C virus (HCV). It was found to inhibit the 2A protease of EV-D68 in cell culture [102]. However, telaprevir in combination treatment carries a box warning for serious skin reactions [129]. Telaprevir was removed from the market as an HCV treatment following the introduction of newer HCV NS3/NS4A protease inhibitors. These newer inhibitors have also demonstrated antiviral activity against multiple viruses, including EV-A71, suggesting potential broader applications of this drug class beyond HCV treatment [130].

#### 6.2.2. Guadinine Hydrochloride and Fluoxetine—2C Protein Inhibitors

Guadinine hydrochloride and fluoxetine are the most studied 2C protein inhibitors, effectively blocking viral replication [111,131]. Guanidine has been shown to inhibit multiple EV-D68 strains in cell culture and demonstrated in vivo antiviral activity in the EV-D68 respiratory animal model [96]. Despite these compelling results in preclinical studies, it has not yet progressed to clinical trials.

Fluoxetine (Prozac) is an FDA approved antidepressant that was shown to reduce EV-D68 replication in cell culture [132,133]. Fluoxetine, particularly its S-enantiomer, binds to an allosteric hydrophobic pocket in protein 2C, stabilizing its hexameric form and inhibiting ATPase activity [94]. It demonstrated potential antiviral effects against EV-D68 in vitro, but it was ineffective against RV [94]. Additionally, it showed no antiviral efficacy in EV-D68 infection mouse models [134]. However, in one case involving an immunocompromised pediatric patient with chronic enteroviral encephalitis, fluoxetine treatment led to stabilization and improvement [135]. In a retrospective multicenter cohort study, fluoxetine was relatively well-tolerated, but it did not show a positive efficacy signal for improving long-term outcomes in patients with AFM [95].

#### 6.2.3. Rupintrivir and V-7404—3C Protease Inhibitors

Rupintrivir (AG7088), a peptide-mimetic inhibitor targeting the viral 3C protease, has shown potent antiviral activity against all *Enterovirus* species, including various human RV genotypes [136]. In vitro studies also confirm its activity against clinical isolates from the three major EV-D68 clusters [98,109]. However, rupintrivir showed no significant efficacy compared to controls in EV-D68 respiratory and neurological mouse infection models [96]. It has not advanced to later-stage clinical trials and remains unapproved.

A similar analog, V-7404, was developed with improved oral bioavailability and demonstrated potent activity against EV-D68 isolates [100]. It has advanced to clinical trials as a potential treatment for serious EV infections [104].

#### 6.2.4. Azvudine—3D Polymerase Inhibitor

Cytidine analog 2′-deoxy-2′-β-fluoro-4′-azidocytidine, also known as azvudine or FNC, exhibits broad-spectrum antiviral activity against various genotypes, including EV-D68, in neonatal mouse model [137]. EV-D68′s RNA-dependent RNA polymerase (3Dpol) plays a crucial role in viral genome replication within the replication organelles, and antivirals targeting 3Dpol may act by inhibiting transcript polymerization or inducing lethal mutagenesis [34]. Originally developed as an HIV treatment, azvudine was approved by the Chinese authorities in 2022 for use in treating mild to moderate COVID-19 cases [91]. There are no existing data on its efficacy against RVs.

#### 6.2.5. Ribavirin—Guanosine Nucleoside Analog

Ribavirin, a synthetic guanosine nucleoside analog, is an older antiviral medication used for treating chronic Hepatitis C, primarily in combination with interferon-based therapies, and other viruses [138,139]. It is also prescribed for various viral hemorrhagic fevers [99]. It interferes with viral RNA synthesis and mRNA capping, primarily through its active metabolite, ribavirin triphosphate (RTP). RTP inhibits viral RNA-dependent RNA polymerase by binding to the enzyme’s nucleotide binding site, leading to misincorporation of nucleotides and premature termination of viral RNA synthesis. This results in the production of defective virions and an error catastrophe due to increased mutagenesis [140]. Additionally, ribavirin depletes intracellular GTP levels by inhibiting the enzyme inosine monophosphate dehydrogenase (IMPDH), further disrupting viral replication [141]. Ribavirin’s efficacy varies across RV serotypes in in vitro studies [98]. In addition, its clinical utility is limited by inconsistent in vivo results, often due to bioavailability challenges and safety profile (anemia and potential teratogenicity). When combined with pegylated interferon α2a, its antiviral effect improves, as evidenced by accelerated RV RNA clearance in patients with hypogammaglobulinemia [142]. Its mechanism in RV infection aligns with its broader antiviral properties, including RNA synthesis inhibition and immunomodulation, which shifts the host immune response toward a Th1 phenotype, promoting antiviral immunity [139].

#### 6.2.6. Molnupiravir and EIDD-1931

This ribonucleoside analog works by inducing lethal mutagenesis during viral RNA replication, thereby disrupting the replication process of RNA viruses. A recent investigation has examined the potential of Beta-D-N4-hydroxycytidine (Emory Institute for Drug Development, EIDD-1931), the active form of molnupiravir, to inhibit *Enterovirus* genotypes, including EV-D68 [92]. Molnupiravir has also demonstrated its broad-spectrum antiviral activity against numerous RNA viruses in animal models [143]. Molnupiravir, the prodrug of EIDD-1931, has received emergency use authorization by FDA for treating SARS-CoV-2 infections and has been approved in several countries [144]. Further studies are necessary to confirm its safety and efficacy specifically for EV-D68 infections in vivo.

### 6.3. Host Factors Antiviral

Antiviral strategies targeting host proteins focus on disrupting the interactions between viruses and the cellular machinery they hijack for replication. Unlike direct-acting antivirals that target viral proteins, host-targeted antivirals aim to inhibit key host factors essential for the viral life cycle, such as enzymes, receptors, or signaling pathways. This approach offers a broader spectrum of activity and reduces the likelihood of viral resistance, as host proteins are less prone to mutation.

#### 6.3.1. SAMHDI

SAMHD1 (Sterile alpha motif and histidine-aspartic acid domain-containing protein 1) is a protein characterized by an N-terminal nuclear localization signal (NLS), a sterile alpha motif (SAM) domain involved in protein interactions, and a catalytic histidine-aspartic acid (HD) domain responsible for its enzymatic activities [145]. Highly expressed in myeloid cells, including dendritic cells, macrophages, and monocytes, SAMHD1 restricts viral infections by depleting cellular deoxynucleoside triphosphates (dNTPs), thereby inhibiting reverse transcription in retroviruses like HIV-1 [146]. A recent study revealed SAMHD1’s broader antiviral capabilities by inhibiting EVs such as EV-A71 and EV-D68 in vitro through mechanisms independent of its canonical dNTPase or RNase activities [118]. Instead, SAMHD1 disrupts viral assembly by competitively binding to the VP1 capsid protein at the same domain where VP2 would normally attach [118].

#### 6.3.2. Enviroxime and Vemurafenib—Pi4kiiiβ Inhibitor

*Enterovirus* genome replication occurs in replication organelles, and host proteins associated with these organelles have been identified as potential antiviral drug targets. Enviroxime inhibits the enzymatic activity of PI4KIIIβ and has been shown to effectively inhibit several strains of EV-D68 [109,111,147]. PI4KIIIβ catalyzes the synthesis of phosphatidylinositol 4-phosphate (PI4P) from phosphatidylinositol (PI), playing a central role in membrane trafficking, Golgi complex integrity, and organelle identity [148]. Enviroxime advanced to clinical trials for treating the common cold caused by human rhinovirus but was discontinued in Phase II due to toxicity concerns [149].

Vemurafenib, an FDA-approved kinase inhibitor developed for the treatment of non-rescuable melanoma, has been repurposed to inhibit PI4KIIIβ, effectively blocking viral replication across EV-A, B, and C, as well as RVs [119].

CUR-N399, KRP-A218, and GSK3923868 have shown positive results in phase I trials for their antiviral potential against RV, further supporting the therapeutic relevance of targeting PI4KIIIβ in enterovirus and other viral infections [106,112,113,114,117].

#### 6.3.3. CRT0066101, CRT0066051 and XX-050—Protein Kinase D Inhibitor

Protein Kinase D (PKD) is a serine/threonine kinase that regulates Golgi membrane dynamics and vesicular trafficking-processes that are extensively exploited during viral replication. PKD has been identified as a host factor required for the replication of several picornaviruses, including human rhinovirus [105]. Interestingly, addition of PKD inhibitors, such as CRT0066101, CRT0066051 and XX-050 in infected HeLa cells led to a significant, concentration-dependent reduction in viral genome replication, protein expression, and infectious titers [105]. The antiviral mechanism is thought to disrupt early post-entry steps in replication, rather than viral entry or interferon signaling [105].

#### 6.3.4. Itraconazole—OSBP Inhibitor

Itraconazole has shown efficacy as an antiviral agent against *Enterovirus*, including poliovirus, coxsackievirus, EV-A71, and RVs. Its antiviral activity is attributed to several mechanisms, primarily involving the inhibition of OSBP and OSBP-related protein 4 (ORP4), which disrupts cholesterol trafficking essential for viral replication [116]. In vivo experiments have shown that itraconazole reduces viral replication in mice and decreases inflammation in the respiratory tract, highlighting its potential as both a therapeutic and prophylactic agent against *Enterovirus* infections [115].

#### 6.3.5. DAS181—Inhibitors Targeting the Cell Surface Sialic Acid Receptors

Sialic acid is a nine-carbon sugar molecule that plays a vital role in cell surface interactions, often found attached to glycoproteins and glycolipids. The way sialic acid connects to other sugars is described by its glycosidic linkage, which is denoted using a specific nomenclature. For example, an α2,6 linkage refers to the bond formed between the C2 carbon of sialic acid and the C6 carbon of the adjacent sugar, such as galactose or N-acetylgalactosamine. This linkage is one of several possible configurations, including α2,3 (C2 to C3) and α2,8 (C2 to C8). The type of linkage determines how sialic acid is presented on cell surfaces, influencing its biological functions. For instance, EV-D68 specifically binds to α2,6-linked sialic acids, enabling viral attachment and entry into host cells [150].

DAS181, a sialidase enzyme that cleaves sialic acid, exhibited potency in inhibiting the replication of both historical and contemporary EV-D68 strains in RDcells. In vitro studies showed that it significantly inhibited contemporary EV-D68 strains (USA-MO/18947, USA-MO/18949, and USA-MO/18956) as well as the prototype strain (Fermon). Additionally, DAS181 demonstrated strong antiviral activity against EV-B, EV-C, EV-D, and rhinovirus A and B strains [35,100]. However, recent data suggest that its effectiveness against EV-D68 may vary by strain and cell type, indicating that its antiviral activity in RD cells is cell-type dependent [34]. It has also demonstrated high efficacy against pandemic H1N1, avian influenza, and drug-resistant influenza strains [108]. DAS181 has received fast track and breakthrough therapy designations by the FDA for the treatment of severe lower respiratory tract parainfluenza virus infection in immunocompromised patients [107].

### 6.4. Antibodies

Recent studies have identified monoclonal antibodies (mAbs) isolated from individuals previously infected with EV-D68 that exhibit broad recognition of diverse antigenic variants [121]. These cross-neutralizing mAbs are now being explored as therapeutic interventions for Enterovirus infections, with the goal of administering them to patients to block viral entry, neutralize circulating virions, or enhance immune clearance. This approach takes advantage of the immune system’s ability to generate potent, strain-transcending antibodies, offering a strategy to bypass the challenges posed by rapid viral evolution and antigenic diversity [121].

#### 6.4.1. Human Antibodies

EV68-228 and EV68-159, two human mAbs with demonstrated neutralizing cross-reactivity in vitro, were tested for antiviral efficacy in mouse models with respiratory or AFM-like neurological conditions. Compared to intravenous immunoglobulin, a therapy made from pooled antibodies from the plasma of healthy donors, both mAbs improved survival and protected mice from disease, whether given before or after infection [121]. EV68-228 is in Phase I since June 2024 to evaluate the safety, pharmacokinetics, and optimal dosage in healthy adult volunteers [122]. EV68-159 is not currently in clinical trials.

Other studies have demonstrated that monoclonal antibodies targeting human intercellular adhesion molecule-1 (ICAM-1) can effectively inhibit rhinovirus-induced exacerbations of lung inflammation [66,151]. ICAM-1 serves as a primary receptor for the major group of rhinoviruses and is upregulated on airway epithelial and endothelial cells during inflammatory responses, facilitating leukocyte adhesion and transmigration [152]. The findings support the potential of anti-ICAM-1 monoclonal antibodies as promising interventions for managing rhinovirus-triggered exacerbations of airway inflammation [66].

#### 6.4.2. Non-Human Antibodies

Several studies have shown that sera derived from animal models, whether from live virus infections, inactivated virus formulated with alum, or virus-like particles (VLPs), are capable of neutralizing both homologous and heterologous EV-D68 strains in vitro [153,154]. When passively transferred to animals, these sera protected against symptoms and death resulting from infection with the respective strain [155,156,157].

A study reported the isolation of the VP1-specific A6-1 mAb from an EV-D68-infected rhesus macaque, demonstrating its ability to neutralize the virus. At low concentrations, these mAbs effectively neutralized EV-D68 Kunming and Fermon strains in vitro [158]. In intranasally infected suckling mice, they provided cross-species protection, aiding in viral clearance and preventing hemorrhage and inflammatory cell aggregation in the lungs and brain. These findings suggest that the A6-1 mAb inhibits EV-D68 intranasal infections and mitigates pathological effects in infected mice [120].

An experimental approach involved sequentially injecting two female BALB/c mice with DNA plasmids encoding capsid proteins and two proteases from various RV-A types, followed by a final boost with the whole virus. This method generated hybridomas expressing mAbs, three of which displayed cross-reactivity against multiple RV-A strains [159]. While RV-A15-specific mAbs exhibited neutralizing activity against RV-A15, the cross-reactive mAbs did not. However, one cross-reactive mAb demonstrated significant antibody-dependent cellular phagocytosis (ADCP) activity. These findings suggest that this approach can produce VP1-specific, cross-reactive antibodies, particularly those with ADCP activity, which may contribute to protection against RV infections. To our knowledge, there are no non-human mAbs currently in clinical trials.

### 6.5. Vaccines

In addition to antiviral treatments, vaccines are a critical area of focus for preventing complications associated with enteroviral infections. The success of the poliovirus vaccine illustrates the significant impact of vaccination against these viruses. Since its introduction in the 1950s, global immunization efforts have substantially reduced polio incidence, bringing the disease close to eradication [160]. While no vaccine is currently approved for EV-D68 or RV, a novel bivalent vaccine targeting EV-A71 and EV-D68 has been developed using formalin-inactivated viruses and polysaccharides from the fungi *Ganoderma lucidum* as an adjuvant. This vaccine demonstrated strong mucosal and systemic immune responses in mice, offering protection against lethal challenges [161]. The vaccine remains at the preclinical stage, focusing on animal models to evaluate efficacy and safety. In addition, antibodies generated against the poliovirus vaccine have demonstrated some cross-reactivity with other *Enterovirus* species, including EV-D68 and EV-A71. This cross-protection remains limited and not universal due to the extensive genetic diversity and numerous serotypes of the genus [50,162]. Preclinical studies report that several experimental RV vaccines induce strong, cross-reactive cellular and humoral immune responses in mice enhancing virus clearance, including the induction of broadly neutralizing antibodies against multiple RV serotypes in rhesus macaques [67,163]. For RV, APL-10456, a prophylactic RV vaccine developed by Apollo Therapeutics Ltd. (patent GB202202738D0), is undergoing IND-enabling to prevent exacerbation of major chronic lung conditions [164,165]. Continued innovation in vaccine design and adjuvant selection will be essential to overcome these challenges and achieve broad, durable protection.

## 7. Conclusions

The recent pandemic has revealed the staggering economic cost of insufficient preparedness for emerging viral threats and the critical importance of proactive research in virology [166]. EVs and RVs continue to evolve, posing a risk as demonstrated by the EV-D68 outbreak in 2014. While no treatment is currently available, many compounds have shown potential. These compounds target a broad range of viral processes, with a particular focus on viral attachment, RO formation, and RNA replication. However, other stages of the viral life cycle have not been explored as potential therapeutic targets. For instance, inhibitors targeting viral release have been successfully developed against HIV, an approach not widely investigated for *Enterovirus* [167,168]. Given these gaps, repurposing existing antivirals developed for other viruses presents an opportunity to explore novel mechanisms of inhibition. Building on the existing cellular, tissue, and animal models that provide a strong foundation for antiviral research, we can anticipate significant advancements in the development of novel therapeutics. By prioritizing investment in these areas today, we can ensure that we are better equipped to address the significant health burden caused by EV-D68, RVs, ultimately improving public health and reducing societal disruption.

## Figures and Tables

**Figure 1 idr-17-00061-f001:**
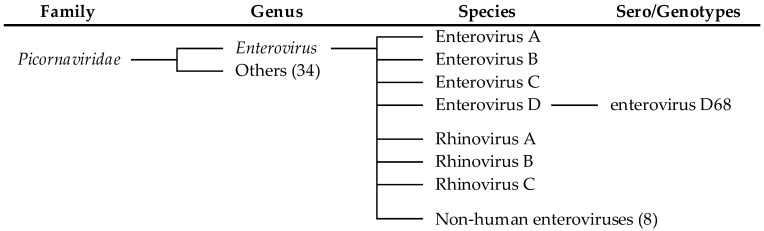
Classification of the EV and RV species linked to illnesses in humans.

**Figure 2 idr-17-00061-f002:**
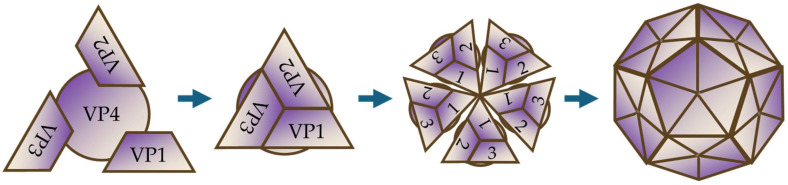
Schematic of *Enterovirus* capsid organization. VP1–VP3 subunits, represented as a triangular unit with VP4 underneath, are assembled into pentameric structures to form a complete icosahedral capsid composed of 60 copies of the VP1–VP4 subunits.

**Figure 3 idr-17-00061-f003:**
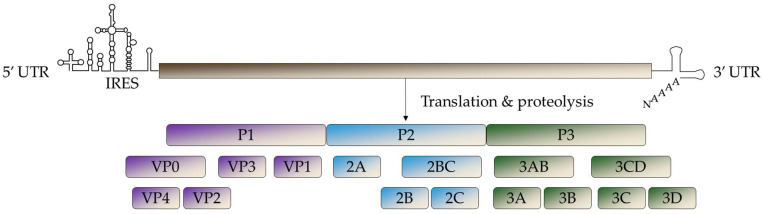
Schematic of *Enterovirus* translation and proteolysis. The genome is translated into a single polyprotein (P1 to P3) then proteolyzed into four structural proteins and seven non-structural proteins. IRES: internal ribosome entry site. UTR: untranslated region.

**Figure 4 idr-17-00061-f004:**
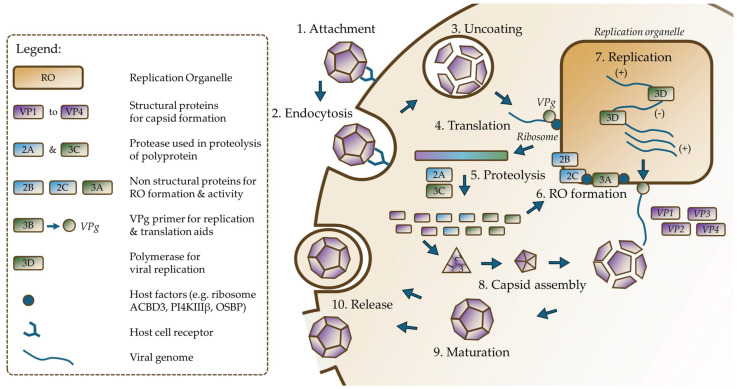
*Enterovirus* replicative cycle. The virus attaches to a host receptor (1) and is subsequently internalized (2). A change in pH triggers the uncoating of the virus (3), resulting in the release of its genome. This RNA genome is translated (4) into a single polyprotein, which undergoes proteolysis (5) to produce four structural proteins and seven non-structural proteins. Replication organelles are generated (6), providing a site for RNA replication (7). Newly replicated genomes are assembled with translated structural proteins (8) to form a viral particle. Upon maturation (9), the newly formed viral particle is released (10) from the host cell. RO = replication organelle.

**Figure 5 idr-17-00061-f005:**
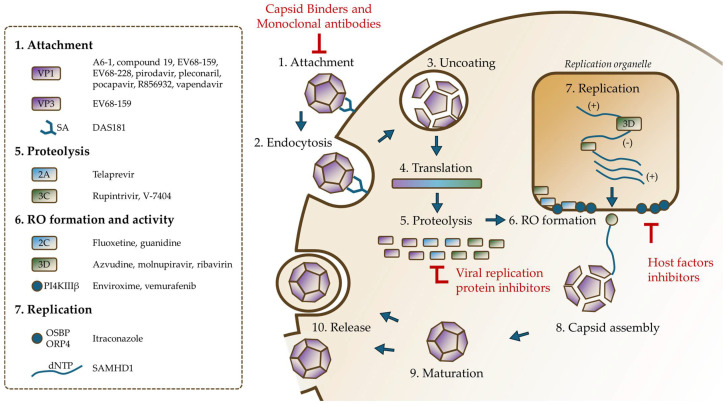
Antiviral candidates targeting EV-D68 and rhinovirus at distinct stages of the viral replicative cycle. Capsid-binding agents and monoclonal antibodies block viral attachment by competitively inhibiting receptor-virus interactions. Viral replication protein inhibitors disrupt polyprotein processing and replication organelle biogenesis by targeting non-structural proteins. Host factor inhibitors impair viral attachment, RO formation, or RNA replication. Each class of inhibitors interferes with critical junctures of the viral life cycle, offering complementary strategies for therapeutic intervention. PI4KIIIβ = PI 4-kinase-IIIβ, SA = Sialic acid, OSBP = oxysterol-binding protein, ORP4 = OSBP-related protein 4.

**Table 1 idr-17-00061-t001:** Antiviral candidates against EV-D68 and RVs.

Drug Class	Antiviral Compound	Virus Targeted	Protein Targeted	Stage of Development	References
Capsid Binders					
	Pleconaril	RV, EV-D68	VP1	Phase II clinical trial for RV, Preclinical for EV-D68	[79,80,81]
	Pirodavir	RV	VP1	Phase II clinical trial	[82,83]
	Pocapavir (V-073)	RV	VP1	Preclinical for RV and EV-D68, available for use in emergency treatment of severe neonatal EV-B infections	[84,85,86]
	Quinoline derivative: Compound 19	EV-D68	Hydrophobic pocket of VP1	Preclinical	[87]
	R856932 (Tetrazole based)	EV-D68	VP1	Preclinical	[88]
	Vapendavir	RV	VP1	Phase II clinical trial	[89,90]
Viral replication protein inhibitors					
	Azvudine (FNC *)	EV-D68	3D	Approved in China for indication other than EV or RV infection	[91]
	EIDD-1931, active form of molnupiravir	EV-D68	3D	EUA ** by FDA, approved in some countries for other infection than EV and RV	[92,93]
	Fluoxetine	EV-D68	2C	FDA approved for indication other than EV or RV infection	[94,95]
	Guanidine	EV-D68	2C	Preclinical	[96]
	Ribavirin	RV	3D	FDA approved for indication other than EV or RV infection	[97,98,99]
	Rupintrivir (AG7088)	RV, EV-D68	3C	Phase II for RV	[100,101]
	Telaprevir	EV-D68	2A	FDA approved for indication other than EV or RV infection	[102,103]
	V-7404	EV-D68	3C	Phase I	[104]
Host factors inhibitors					
	CRT0066101, CRT0066051 and XX-050	RV	Protein Kinase D (PKD)	Preclinical	[105]
	CUR-N399	RV	PI4KIIIβ	Phase I	[106]
	DAS181	RV, EV-D68	Sialic acid	Fast Track and Breakthrough Therapy designations by the FDA for indications other than EV or RV infection	[100,107,108]
	Enviroxime	RV, EV-D68	PI4KIIIβ	Phase II for RV, preclinical for EV-D68	[109,110,111]
	GSK3923868	RV	PI4KIIIβ	Phase I	[112,113,114]
	Itraconazole	RV	OSBP, ORP4	FDA approved for indication other than EV or RV infection	[115,116]
	KRP-A218	RV	PI4KIIIβ	Phase I	[117]
	SAMHD1 ***	EV-D68	Cellular deoxynucleotide triphosphate (dNTP)	Preclinical	[118]
	Vemurafenib	RV	PI4KIIIβ	FDA approved for indication other than EV or RV infection	[119]

Antibodies					
	A6-1	EV-D68	VP1	Preclinical	[120]
	EV68-159	EV-D68	VP1, VP3	Preclinical	[121]
	EV68-228	EV-D68	VP1	Phase 1	[121,122]
	14C11	RV	ICAM-1	Preclinical	[66]

* 2′-deoxy-2′-β-fluoro-4′-azidocytidine, ** Emergency Use Authorization, *** Sterile alpha motif and histidine-aspartic acid domain-containing protein 1.

## Data Availability

No new data were generated or analyzed.

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
