# Peer review of "Advances in the Treatment of Enterovirus-D68 and Rhinovirus Respiratory Infections"

_2036-7449, 2025, doi:10.3390/idr17030061_

Round 1
Reviewer 1 Report
Comments and Suggestions for Authors
The present paper, titled "Advances in the Treatment of Enterovirus-D68 and Rhinovirus Respiratory Infections," is a well-written review that includes the classification, biology, genome organization, and replication of Enterovirus and rhinovirus, with emphasis on current strategies in Enterovirus antiviral research. The authors describe all potential antiviral compounds that aim to disrupt virus replication and spread, usually targeting viral proteins or essential host factors. The figures and table included are very informative. The authors maintain a clear central theme throughout the article, making it engaging and easy to read.
The paper is of great interest to scientists working in the field of enterovirus D-68 and rhinoviruses.
Page 5; Lines 149-150: authors wrote: "EV-D68 exemplifies these concerns, having caused a large epidemic in North America in 2014." What about other regions? Like Europe? Australia? Asia?Author Response
We would like to thank Reviewer 1 for taking the time to review our manuscript and for the insightful comments provided.
Comment 1: Page 5; Lines 149-150: authors wrote: "EV-D68 exemplifies these concerns, having caused a large epidemic in North America in 2014." What about other regions? Like Europe? Australia? Asia?
Response 1: We added information about other regions (see lines 154-155).
“EV-D68 exemplifies these concerns, having caused a large epidemic in North America in 2014, with concurrent outbreaks and sporadic cases also reported in Europe, Asia, Australia, and other regions, highlighting its global public health relevance [43-49].”
Reviewer 2 Report
Comments and Suggestions for Authors
There is no discussion of broad spectrum anti-viral approaches such as type I and type III IFNs, and innate immune stimulators. There are many such approaches, several with good evidence of activity. Either discuss or mention their existence, but say you deliberately choose to omit.
Specific comments:
- Line 35, ref 4: Citation incomplete, give details (URL).
- Line 54: “Antiviral approaches”.
- Line 63: “correlate strongly with serotype designations”. May be true for the enterovirus B species - what about other EVs and RVs?
- Line 93: “contains”.
- Line 96: “P2 region”. Refer to Fig 3, as P2 not mentioned before.
- 3: P1 & P2 should each extend to the junction between VP1 and 2A.
- Lines 167-179: This is all very EV related and does not mention RV mouse models of RV infection (PMID: 18246079) and of RV-induced exacerbation of lung diseases (PMID: 25783022 & PMID: 28459437) and RV antiviral (PMID: 23935498) and vaccine work (PMID: 24086140) in such mouse models. Please include.
- Line 192: “has” instead of “have”.
- Table 2: “Vapendavir”. Update with NCT06149494, results are expected in summer of 2025.
- Table 2: “Viral replication protein inhibitors”
- Table 2, Ref. 91. Citation incomplete, please complete.
- Tabel 2, “Host factors inhibitors”. Update with NCT05016687, NCT04908800 and NCT05677347, NCT04585009 & NCT05398198.
- Table 2: “monoclonal antibodies”. Add PMID: 23935498
- Lines 244-248: Update as above with NCT06149494, results are expected in summer of 2025.
- Line 260: “cell model”.
- Line 264. “with EC50 values”. High, low, medium? Please provide.
- Line 268: “rat models”.
- Line 274: “Telaprevir is an antiviral”.
- Line 346, title: “Host Factor antiviral approaches”. Add a section on PKD (PMID: 28228588).
- Line 365, title: “inhibitors”.
- Lines 366- 376: Update with NCT05016687, NCT04908800 and NCT05677347, NCT04585009 & NCT05398198.
- Line 398: “Rhabdomyosarcoma (RD)”. Already abbreviated.
- Line 401: “rhinovirus A and B strains”.
- Line 408, paragraph 6.4: Update with PMID: 23935498.
- Line 417, title: “Human monoclonal antibodies”. Pooled human Abs are not monoclonal Abs. Delete/rephrase.
- Line 218 and 421: “mAbs”. Pooled human Abs are not monoclonal Abs. Delete/rephrase.
- Line 426, title: “monoclonal”. Sera are not monoclonal Abs. Delete/rephrase.
- Lines 427-438. Sera are not monoclonal Abs. Delete/rephrase.
- Lines 449-463: “Vaccines”. Please include RV vaccine work (PMID: 24086140 and PMID: 27653379).. Also patent GB202202738D0, Apollo Therapeutics Ltd have AP-10456 which is undergoing IND enabling studies - see https://www.apollotx.com/pipeline/. Several abstracts at ERS and ATS report cross reactive immune responses see: https://publications.ersnet.org/content/erj/64/suppl68/oa5461
- Line 465: replace “Previous” with “the recent”.
- Line 478: “when EV-D68 or RVs or similar viruses pose a pandemic threat…” RVs and EVs pose a threat to health now, and they are most unlikely to cause a pandemic. Suggest re-phrase.
Author Response
We would like to thank Reviewer 2 for the thoughtful assessment of our manuscript.
Comment 1: There is no discussion of broad spectrum anti-viral approaches such as type I and type III IFNs, and innate immune stimulators. There are many such approaches, several with good evidence of activity. Either discuss or mention their existence, but say you deliberately choose to omit.
Response 1: We have included a mention of broad-spectrum antiviral strategies, including type I and type III interferons as examples of innate immune stimulators, in the revised version of the manuscript (see lines 191-201):
“Broad-spectrum antiviral approaches aim to target a wide range of viruses by interfering with essential viral or host processes required for replication, rather than focusing on a single virus or viral family [72]. These strategies include both direct-acting antivirals and host-targeted therapies [73]. Examples of the former include nucleoside analogues like ribavirin, which inhibit viral polymerases across multiple RNA and DNA viruses, and favipiravir, which targets the RNA-dependent RNA polymerase of various RNA viruses [74,75]. Host-directed approaches include type I and type III interferons (IFNs), which stimulate the innate immune response by binding to their specific receptors and activating the JAK-STAT signaling pathway. This leads to the expression of hundreds of interferon-stimulated genes (ISGs) that induce antiviral states in infected and neighboring cells [76].”
Comment 2: Line 35, ref 4: Citation incomplete, give details (URL).
Response 2: We have made the correction in reference 4, which can be read as follow:
“4. ICTV Report Consortium Picornaviridae Available online: https://ictv.global/report/chapter/picornaviridae/picornaviridae/enterovirus (accessed on 23 February 2025).”
Comment 3: Line 54: “Antiviral approaches”.
Response 3: The correction has been made.
Comment 4: Line 63: “correlate strongly with serotype designations”. May be true for the enterovirus B species - what about other EVs and RVs?
Response 4: Mention of correlation between serotype and genetic designation was removed.
Comment 5: Line 93: “contains”.
Response 5: The correction has been made.
Comment 6: Line 96: “P2 region”. Refer to Fig 3, as P2 not mentioned before.
Response 6: The figure 3 is mentioned at the beginning of the same paragraph (see line 89).
Comment 7: 3: P1 & P2 should each extend to the junction between VP1 and 2A.
Response 7: Figure 3 has been modified accordingly (see line 98).
Comment 8: Lines 167-179: This is all very EV related and does not mention RV mouse models of RV infection (PMID: 18246079) and of RV-induced exacerbation of lung diseases (PMID: 25783022 & PMID: 28459437) and RV antiviral (PMID: 23935498) and vaccine work (PMID: 24086140) in such mouse models. Please include.
Response 8: The information on RV mouse models, including studies on infection, lung disease exacerbation, antiviral treatments, and vaccine development, has been added to the text (see line 176-178):
“Rodent models, particularly mice, have been extensively used to study Enterovirus infection, viral-induced exacerbation of lung diseases, and for testing antivirals and vaccines [64–68].”
Comment 9: Line 192: “has” instead of “have”.
Response 9: The correction has been made.
Comment 10: Table 2: “Vapendavir”. Update with NCT06149494, results are expected in summer of 2025.
Response 10: The reference was added.
Comment 11: Table 2: “Viral replication protein inhibitors”
Response 11: The correction has been made.
Comment 12: Table 2, Ref. 91. Citation incomplete, please complete.
Response 12: The citation previously listed as Ref. 91 (now Ref. 92) has been completed.
Comment 13: Table 2, “Host factors inhibitors”. Update with NCT05016687, NCT04908800 and NCT05677347, NCT04585009 & NCT05398198.
Response 13: We have updated Table 2 to include the corresponding compounds and trials.
Comment 14: Table 2: “monoclonal antibodies”. Add PMID: 23935498
Response 14: The anti ICAM-1 antibody was added to the table.
Comment 15: Lines 244-248: Update as above with NCT06149494, results are expected in summer of 2025.
Response 15: The text has been updated to include clinical trial NCT06149494, with a note that results are expected in summer 2025 (see lines 269–270).
“Although both drugs remain experimental, results from a phase II clinical trial of vapendavir are expected in summer 2025 [84,90,91].”
Comment 16: Line 260: “cell model”.
Response 16: The correction has been made.
Comment 17: Line 264. “with EC50 values”. High, low, medium? Please provide.
Response 17: We have clarified in the text that the compounds discussed have ECâ‚…â‚€ values ranging from 0.05 to 0.10 μM. (see line 286).
“This compound exhibited broad-spectrum antiviral effects in vitro, with an EC50 value ranging from 0.05 to 0.10 μM against various EV-D68 various clinical EV-D68 strains as well as the prototype strain in RD cell lines.”
Comment 18: Line 268: “rat models”.
Response: 18: The correction has been made.
Comment 19: Line 274: “Telaprevir is an antiviral”.
Response 19: The correction has been made.
Comment 20: Line 346, title: “Host Factor antiviral approaches”. Add a section on PKD (PMID: 28228588).
Response 20: A section on PKD, including the information from PMID: 28228588, has been added to the “Host Factor Antiviral Approaches” section (see lines 402-410).
“6.3.3. CRT0066101, CRT0066051 & XX-050 - Protein Kinase D Inhibitor
Protein Kinase D (PKD) is a serine/threonine kinase that regulates Golgi membrane dynamics and vesicular trafficking-processes that are extensively exploited during viral replication. PKD has been identified as a host factor required for the replication of several picornaviruses, including human rhinovirus (HRV) [106]. Interestingly, the addition of PKD inhibitors, such as CRT0066101, CRT0066051 & XX-050 in infected HeLa cells led to a significant, concentration-dependent reduction in viral genome replication, protein expression, and infectious titers [106]. The antiviral mechanism is thought to disrupt early post-entry steps in replication, rather than viral entry or interferon signaling [106].”
Comment 21: Line 365, title: “inhibitors”.
Response 21: The correction has been made.
Comment 22: Lines 366- 376: Update with NCT05016687, NCT04908800 and NCT05677347, NCT04585009 & NCT05398198.
Response 22: The text has been modified accordingly to incorporate this information (Line 399-401).
“CUR-N399, KRP-A218, and GSK3923868 have shown positive results in phase I trials for their antiviral potential against RV, further supporting the therapeutic relevance of targeting PI4KIIIβ in enterovirus and other viral infections [107,113–115,118].”
Comment 23: Line 398: “Rhabdomyosarcoma (RD)”. Already abbreviated.
Response 23: The correction has been made.
Comment 24: Line 401: “rhinovirus A and B strains”.
Response 24: The correction has been made.
Comment 25: Line 408, paragraph 6.4: Update with PMID: 23935498.
Response 25: The text has been updated to incorporate the findings from PMID: 23935498, which support the use of anti-ICAM-1 monoclonal antibodies to inhibit rhinovirus-induced airway inflammation (see line 460-466).
“Other studies have demonstrated that monoclonal antibodies targeting human intercellular adhesion molecule-1 (ICAM-1) can effectively inhibit rhinovirus-induced exacerbations of lung inflammation [67,153]. ICAM-1 serves as a primary receptor for the major group of rhinoviruses and is upregulated on airway epithelial and endothelial cells during inflammatory responses, facilitating leukocyte adhesion and transmigration [154]. The findings support the potential of anti-ICAM-1 monoclonal antibodies as promising interventions for managing rhinovirus-triggered exacerbations of airway inflammation [67].”
Comment 26: Line 417, title: “Human monoclonal antibodies”. Pooled human Abs are not monoclonal Abs. Delete/rephrase.
Comment 27: Line 218 and 421: “mAbs”. Pooled human Abs are not monoclonal Abs. Delete/rephrase.
Comment 28: Line 426, title: “monoclonal”. Sera are not monoclonal Abs. Delete/rephrase.
Comment 29: Lines 427-438. Sera are not monoclonal Abs. Delete/rephrase.
Response for 26 to 29: Mentions of “monoclonal” in section titles have been removed to reflect the content accurately.
Comment 30: Lines 449-463: “Vaccines”. Please include RV vaccine work (PMID: 24086140 and PMID: 27653379). Also patent GB202202738D0, Apollo Therapeutics Ltd have AP-10456 which is undergoing IND enabling studies - see https://www.apollotx.com/pipeline/. Several abstracts at ERS and ATS report cross reactive immune responses see: https://publications.ersnet.org/content/erj/64/suppl68/oa5461
Response 30: The text has been modified accordingly to incorporate this information (see line 506-513).
“Preclinical studies report that several experimental RV vaccines induce strong, cross-reactive cellular and humoral immune responses in mice, enhancing virus clearance, including the induction of broadly neutralizing antibodies against multiple RV serotypes in rhesus macaques [165,166]. For RV, APL-10456, a prophylactic RV vaccine developed by Apollo Therapeutics Ltd (patent GB202202738D0), is undergoing IND-enabling to prevent exacerbation of major chronic lung conditions [167,168]. Continued innovation in vaccine design and adjuvant selection will be essential to overcome these challenges and achieve broad, durable protection.”
Comment 31: Line 465: replace “Previous” with “the recent”.
Response 31: The correction has been made.
Comment 31: Line 478: “when EV-D68 or RVs or similar viruses pose a pandemic threat…” RVs and EVs pose a threat to health now, and they are most unlikely to cause a pandemic. Suggest re-phrase.
Response 31: The sentence has been revised to reflect this comment (see lines 528-530).
“By prioritizing investment in these areas today, we can ensure that we are better equipped to address the significant health burden caused by EV-D68, RVs, ultimately improving public health and reducing societal disruption.”
Reviewer 3 Report
Comments and Suggestions for Authors
A Medline search shows no comprehensive reviews of rhinovirus or EV-68 therapeutics for at least fours years and none combined. The combination is useful drawing attention to importance of both for asthma treatment where there could be very significant and targeted application for exacerbation prone children. In general, it is a very readable and useful for both the therapeutics and the targets for therapies. It has a good account of synthetic inhibitors.
Although interferon is mentioned in some contexts the review does not give much of an account of its use.
The problems of the mouse models could be better explained. At least historically they are not very useful for rhinovirus even for RV receptor producing transgenics. The literature shows the use of hSCARB2 transgenic mice for enterovirus 71 and coxsackie virus 16. They do not appear to have been used for Ev-D68. Is that possible? Are there now reliable in vitro cultures systems for the virulent rhinovirus C species ?
Author Response
We thank Reviewer 3 for their careful review of our manuscript.
Comment 1. Although interferon is mentioned in some contexts the review does not give much of an account of its use.
Response 1: In response to this comment, we have expanded upon the use of interferon, providing additional context regarding its therapeutic role (see lines 191-201).
“Broad-spectrum antiviral approaches aim to target a wide range of viruses by interfering with essential viral or host processes required for replication, rather than focusing on a single virus or viral family [72]. These strategies include both direct-acting antivirals and host-targeted therapies [73]. Examples of the former include nucleoside analogues like ribavirin, which inhibit viral polymerases across multiple RNA and DNA viruses, and favipiravir, which targets the RNA-dependent RNA polymerase of various RNA viruses [74,75]. Host-directed approaches include type I and type III interferons (IFNs), which stimulate the innate immune response by binding to their specific receptors and activating the JAK-STAT signaling pathway. This leads to the expression of hundreds of interferon-stimulated genes (ISGs) that induce antiviral states in infected and neighboring cells [76].”
Comment 2: The problems of the mouse models could be better explained. At least historically they are not very useful for rhinovirus even for RV receptor producing transgenics. The literature shows the use of hSCARB2 transgenic mice for enterovirus 71 and coxsackie virus 16. They do not appear to have been used for Ev-D68. Is that possible? Are there now reliable in vitro cultures systems for the virulent rhinovirus C species ?
Response 2: We have expanded the section on animal models to better explain the limitations of mouse models, particularly in the context of rhinovirus research (see lines 176-178).
“Mouse models for rhinovirus (RV) infection have historically been challenging due to species-specific receptor differences [61]. However, recent advances have enabled the development of more useful models.”
You are correct that hSCARB2 transgenic mice have not been used for EV-D68. We have revised the text to remove this example and instead included ICAM-1 transgenic mice, which are more relevant to the viruses discussed in this review.
We have added two references supporting the development of in vitro culture systems for RV-C (line 173).
"57. Basta, H.A.; Ashraf, S.; Sgro, J.-Y.; Bochkov, Y.A.; Gern, J.E.; Palmenberg, A.C. Modeling of the Human Rhinovirus C Capsid Suggests Possible Causes for Antiviral Drug Resistance. Virology 2014, 448, 82–90, doi:10.1016/j.virol.2013.10.004.
58. Mello, C.; Aguayo, E.; Rodriguez, M.; Lee, G.; Jordan, R.; Cihlar, T.; Birkus, G. Multiple Classes of Antiviral Agents Exhibit In Vitro Activity against Human Rhinovirus Type C. Antimicrob. Agents Chemother. 2014, 58, 1546–1555, doi:10.1128/AAC.01746-13.”
Reviewer 4 Report
Comments and Suggestions for Authors
The manuscript reviews current treatments for enteroviruses and rhinoviruses. The manuscript succinctly describes different antivirals, monoclonal antibodies, and vaccines that have been tested for efficacy against either enteroviruses or rhinoviruses. Overall, the manuscript provides a useful review of the current state of antiviral development of these two viruses. Just some minor corrections:
- Line 143: Can the authors expand on this timeline? Is there any information on how long it takes for various viral proteins to appear?
- Line 274: The first sentence of this paragraph is not a proper sentence --- there's no verb.
- Line 327: should be "serotypes in in vitro studies"
- Line 342: remove "for" at end of line
- Line 343: missing space between several and countries.
Author Response
We thank Reviewer 4 for their careful reading of the manuscript and this constructive suggestion.
Comment 1: Line 143: Can the authors expand on this timeline? Is there any information on how long it takes for various viral proteins to appear?
Response 1: We have expanded the text to provide more detail on the timeline of enterovirus replication and protein expression (lines 144-147).
“The virus replication cycle generates virions within a few hours of the initial infection, with viral RNA replication beginning 2-3 hours post-infection and translation starting shortly after at 3-4 hours [28,40]. Most viral proteins reach high levels between 4–7 hours post-infection, and new virions are typically released within 8–10 hours, completing the efficient replication cycle of enteroviruses [41,42].”
Comment 2: Line/ 274: The first sentence of this paragraph is not a proper sentence --- there's no verb.
Response 2: The correction has been made (line 296).
“Telaprevir is an antiviral originally developed to treat Hepatitis C virus (HCV).”
Comment 3: Line 327: should be "serotypes in in vitro studies"
Response 3: The correction has been made (line 349).
Comment 4: Line 342: remove "for" at end of line
Response 4: The correction has been made (line 364).
Comment 5: Line 343: missing space between several and countries.
Response 5: The correction has been made (line 365).